# Combined Neuroendocrine Carcinoma and Hepatocellular Carcinoma of the Liver: Systematic Literature Review Suggests Implementing Biological Characterization to Optimize Therapeutic Strategy

**DOI:** 10.3390/cancers17071074

**Published:** 2025-03-22

**Authors:** Daniela Sambataro, Sandro Bellavia, Paolo Di Mattia, Danilo Centonze, Carmela Emmanuele, Annalisa Bonasera, Giuseppe Caputo, Andrea Maria Onofrio Quattrocchi, Ernesto Vinci, Vittorio Gebbia, Maria Rosaria Valerio

**Affiliations:** 1Medical Oncology Unit, Umberto I Hospital, 94100 Enna, Italy; annalisa.bonasera@asp.enna.it (A.B.); giuseppe.caputo@asp.enna.it (G.C.); andrea.quattrocchi@asp.enna.it (A.M.O.Q.); ernesto.vinci@asp.enna.it (E.V.); 2Department of Medicine and Surgery, Kore University, 94100 Enna, Italy; paolo.dimattia@unikore.it (P.D.M.); vittorio.gebbia@unikore.it (V.G.); 3Pathology Unit, Umberto I Hospital, 94100 Enna, Italy; sandro.bellavia@asp.enna.it (S.B.); carmela.emmanuele@asp.enna.it (C.E.); 4Surgery Unit, Umberto I Hospital, 94100 Enna, Italy; danilo.centonze@asp.enna.it; 5Medical Oncology Unit, Policlinic, University of Palermo, 90127 Palermo, Italy; mariarosaria.valerio@unipa.it

**Keywords:** MINEN, neuroendocrine, hepatocellular, biopsy, liver

## Abstract

Mixed neuroendocrine–non-neuroendocrine tumors (MINEN) of the liver, particularly those involving hepatocellular carcinoma (HCC) and neuroendocrine carcinoma (NEC), are extremely rare. Our review of 45 cases highlights clinical characteristics, treatment strategies, and outcomes. Treatment options are limited, and survival remains poor. Understanding the biology of these tumors through molecular studies may help develop better treatment strategies. Future research should focus on novel therapies, including immunotherapy, to improve outcomes in this challenging disease.

## 1. Introduction

Neuroendocrine neoplasms (NENs) comprise a heterogeneous group of tumors arising from neuroendocrine cells. They most commonly occur in the gastroenteropancreatic (GEP) and bronchopulmonary systems. Although historically considered rare, their incidence has been steadily increasing, likely due to greater disease awareness and advancements in diagnostic techniques, particularly in endoscopy and nuclear medicine. A population-based study has reported a 6.4-fold increase in incidence [1].

NENs are broadly classified into two major categories: well-differentiated neuroendocrine tumors (NETs) and poorly differentiated neuroendocrine carcinomas (NECs). NETs generally follow an indolent course, whereas NECs exhibit aggressive clinical behavior. These neoplasms are often associated with characteristic clinical features such as flushing, diarrhea, and cardiac manifestations, along with site-specific symptoms, including bronchospasm, myopathy, skin pigmentation, and other paraneoplastic syndromes.

NECs are characterized by a poorly differentiated morphology, displaying diffuse solid growth, extensive necrosis, and marked cytological atypia. They are further subclassified into small-cell and large-cell NECs.

In epithelial tumors of most organs, including those of the gastrointestinal tract, cells exhibiting both neuroendocrine and non-neuroendocrine features can coexist. The biological behavior of such neoplasms remains a topic of ongoing debate. Mixed tumors were first described by Cordier [2] in the early 20th century as neoplasms of the gastrointestinal tract containing both adenocarcinoma and neuroendocrine components. Today, the term mixed neuroendocrine–non-neuroendocrine neoplasms (MiNENs) refers to a heterogeneous group of rare GEP tumors that pose significant diagnostic and therapeutic challenges for both clinicians and pathologists [3,4].

These neoplasms are defined by the coexistence of neuroendocrine (NE) and non-NE components, each comprising at least one-third of the tumor mass.

Data from the Surveillance, Epidemiology, and End Results (SEER) database (2000–2019) indicate that GEP-MiNENs are associated with a worse prognosis compared to pure NECs, with an increasing incidence over time. Several independent prognostic factors have been identified, including surgical intervention, tumor size, and the presence of lymph node or distant metastases [5]. An Italian study analyzing 51 cases of advanced GEP-MiNENs (32 colorectal, 14 gastroesophageal, and 5 pancreatobiliary) demonstrated that the NEC component plays a dominant role in disease progression and overall prognosis [6].

However, hepatic MiNENs with coexistent hepatocellular (HCC) and NEC remain exceptionally rare, with only a few documented cases in the literature. This review aims to identify key features that may enhance the diagnosis and treatment strategies for this exceedingly rare entity.

## 2. Materials and Methods

Published studies on the patient cases of primary, coexistent, liver HCC and NEC were identified through a systematic literature search on PubMed using the keywords “neuroendocrine”, “hepatocellular carcinoma”, “MINEN”, “liver”, and “hepatic”, covering all available studies up to January 2025.

Records identified by the searches were screened according to Preferred Reporting Items for Systematic Reviews and Meta-Analyses (PRISMA) guidelines [7], and the review was not registered (Figure 1).

Full papers were reviewed by two authors, and uncertainties were resolved by a third reviewer. During data extraction, inherent data were extracted manually for each included report. Eligible studies were exported using Zotero Software v6.0.26 (Zotero.org). Overall, 845 records were identified. Only 45 reports reporting well-documented clinical cases pertinent to this review were included [8,9,10,11,12,13,14,15,16,17,18,19,20,21,22,23,24,25,26,27,28,29,30,31,32,33,34,35,36,37,38,39,40,41,42,43,44,45,46,47,48,49,50,51,52] (Table 1). Among them, three reports (one with two cases with three distinct neoplastic components, HCC, CC, and NEC) were described in three abstracts (Japanese, Korean, and Chinese) [22,23,48].

Among these, 45 cases involved mixed HCC and NEC (Table 1). For data analysis, we included a case diagnosed and treated in our hospital, which is the subject of personal communication (Table 1).

For the description of results, we used percentages and medians with range, using only known data.

## 3. Results

Among the 46 cases of mixed HCC and NEC, 90% of patients were male, with a median age of 66.5 years (range: 33–84). The combined type was found in 65% of cases, collision in 35% of cases. Among them, 74% had hepatitis B or C, and liver cirrhosis was observed in 38% of patients. Most cases were observed in Asian populations (Figure 2).

Various types of treatment summarized in Table 2 were used, and most carried out surgery with a median survival of 10 months (range 1–24 months). In most patients treated with chemotherapy, the combination of cisplatin and etoposide was used. A recent record reports the case of a patient treated with atezolizumab and bevacizumab, at progression with lenvatinib and then cabozantinib, with a survival of 24 months [47]. Notably, six patients had undergone prior transarterial chemoembolization (TACE), and one had received radiofrequency ablation (RFA) (Figure 3).

The median overall survival was 10 months (range 1–28; 95% confidence limits of 6–12 months). The mean overall survival was 10.9 months (95% confidence limits of 8.3–13.4 month) with a 75.3% coefficient of variations. These data reflect the heterogeneity of patients and the aggressive nature of these neoplasms.

Additionally, five cases exhibited three distinct tumor components: HCC, NEC, and cholangiocarcinoma (CC) [48,50,51,52], while one case involved NEC and CC in a patient with a prior history of HCC [49].

## 4. Discussion

Most epithelial neoplasms of the gastrointestinal (GI) tract and hepatopancreatobiliary system are classified as either pure glandular (adenocarcinoma), squamous (squamous cell carcinoma), or pure neuroendocrine neoplasms (NENs) [2].

Neuroendocrine neoplasms of the digestive system can arise in various epithelial organs and encompass multiple subtypes with distinct etiologies, clinical presentations, histological characteristics, and prognostic outcomes. Despite site-specific classifications, different classification systems share common features [53].

The most recent WHO classification differentiates well-differentiated neuroendocrine tumors (NETs), previously referred to as carcinoid tumors when located in the GI tract, from poorly differentiated neuroendocrine carcinomas (NECs), which exhibit distinct biological and clinical characteristics [54].

Although both NETs and NECs share neuroendocrine differentiation, as demonstrated by immunolabeling for synaptophysin, chromogranin A, neuron-specific enolase (NSE), and CD56 (NCAM), they exhibit distinct morphological features. NETs typically display organoid architecture (e.g., cords, nests, and ribbons), uniform nuclear features, coarsely stippled chromatin, and minimal or absent necrosis. In contrast, NECs generally present a solid growth pattern with sheets of cells and tightly packed nuclei. Small-cell NECs (SCNECs) have fusiform nuclei with finely granular chromatin, whereas large-cell NECs (LCNECs) display more rounded or atypical nuclei, sometimes with prominent nucleoli. Necrosis is typically extensive [55].

NETs may undergo grade progression (G1 to G3) during tumor evolution, particularly between primary and metastatic sites [56].

NECs, however, originate from precursor cells that typically give rise to non-neuroendocrine carcinomas of the corresponding organ and may contain non-neuroendocrine components such as adenocarcinoma or squamous cell carcinoma [53,54,57].

The mainstay of treatment of NET has been surgery, but with the evolution of pharmacotherapy in recent decades, the horizon of options has broadened. Various medical therapies are currently available, including somatostatin analogs, targeted agents, chemotherapy, peptide receptor radionuclide therapy (PRRT), and immunotherapy. Regarding functional well-differentiated G1-G2 neuroendocrine tumors (NETs) (mitotic rate: ≤20, Ki-67: ≤20%) with liver metastases, a population-based study concluded that resection of the primary tumor improves survival outcomes [58].

Treatment of NEC-GEP is a challenge because of the aggressive behavior of these tumors; they are often diagnosed at an advanced stage and curative treatment is not possible. The most widely used systemic treatment involves platinum-based combination, better biological understanding paves the way for target therapy [59].

Glandular neoplasms, and to a lesser extent squamous neoplasms, may exhibit a minor population of interspersed neuroendocrine cells or partial neuroendocrine differentiation within the same cells. This phenomenon can be identified via immunohistochemical staining but does not influence classification [57].

Less commonly, epithelial neoplasms contain significant proportions of both neuroendocrine and non-neuroendocrine cell populations. In earlier classifications, neoplasms with ≥30% of each component were categorized as “mixed adeno-neuroendocrine carcinoma (MANEC)”. However, the term has been revised to “mixed neuroendocrine–non-neuroendocrine neoplasm (MiNEN)” to acknowledge that the non-neuroendocrine component may not necessarily be adenocarcinoma [3,4].

MiNENs represent a heterogeneous and infrequent group of tumors within the gastroenteropancreatic system, often posing diagnostic and clinical challenges [3,4]. In these tumors, the neuroendocrine (NE) and non-NE components should be morphologically and immunohistochemically distinguishable but clonally related. In the GI tract and hepatopancreatobiliary system, both components are usually carcinomas, with the neuroendocrine component often being a poorly differentiated NEC (LCNEC or SCNEC) [60].

If the neuroendocrine component constitutes less than 30% of the tumor, it may still be mentioned in the diagnosis, particularly if it consists of small-cell NEC (SCNEC), due to its prognostic significance. However, this does not alter the primary classification [53,54]. Generally, MiNENs comprising adenocarcinoma with a NEC component exhibit a poor prognosis, resembling that of pure NECs [2].

Rarely, MiNENs may contain a well-differentiated NET component, but the prognostic implications of such cases remain unclear and warrant further investigation [53]. Whenever feasible, the NE and non-NE components of MiNENs should be graded individually, as emerging evidence suggests that the grade of the NE component correlates with prognosis [3,61]. The NET component is graded as G1, G2, or G3 based on proliferation indices, determined by mitotic count and the Ki-67 index, evaluated by counting at least 500 cells in hotspot regions identified at scanning magnification [62].

Hepatic NENs are typically a solitary, circumscribed parenchymal mass, with an average of 6.5 cm.

Hepatic NEC is a rare occurrence, primarily observed in patients without underlying chronic liver disease [8,63,64,65]. The exact cause of this neoplasm remains mostly unclear [61,62]. Given that NEC commonly metastasizes the liver, it is essential to initially rule out a secondary location [60,61]. Moreover, pathological features consistent with NECs have been observed within HCC nodules [64,65,66].

Hepatic MiNENs are more frequently reported than pure hepatic NENs. Most cases include an HCC component, which may predominate [2]. The NEC component in hepatic MiNENs closely resembles its extrahepatic counterparts, exhibiting high mitotic activity, extensive necrosis, a Ki-67 index > 50%, express synaptophysin, and, less frequently, chromogranin A.

In contrast to the more common finding of HCC plus cholangiocellular carcinoma (CCC) in the liver, the coexistence of HCC and NEC is infrequent, with few reported cases in the literature [67], and no case of HCC mixed with NEC reported in patients with a history of gallbladder cancer, such as in our patient.

Nomura et al. investigated 1235 hepatic tumors, showing an incidence of concurrent HCC and NEC of 0.46% [24]. Most of these cases were associated with viral hepatitis B or C, predominantly affecting male patients with a mean age of 64.6 years (range 43–76). A pooled analysis conducted by Mao et al. reported only 28 cases of mixed HCC and primary neuroendocrine tumor (PHNET) of the liver over a 29-year time interval. Nearly 93% of these patients were male, with a median age of 68. Hepatitis B or C infection and liver cirrhosis were observed in 78% and 35% of patients, respectively. Only three patients underwent TACE. The diagnosis of mixed neoplasm was associated with poor outcomes, with a median survival of only 17.8 months [68]. Bu et al. very recently reported a 58-year-old Chinese male patient positive for hepatitis affected by dual primary liver cancer involving small-cell NEC in segments VII-VIII and HCC in segment V, poorly responsive to chemotherapy, and subsequently treated with TACE and hepatic artery infusion chemotherapy, resulting in partial remission. At progression in the brain, the patients received sorafenib and tirelizumab, a PD-1 immune checkpoint inhibitor, without severe toxicity. The patient survived for 16 months after diagnosis [69].

The origin of pure primary NEC of the liver is still debated, with suggestions about its origin ranging from an ectopic pancreatic tissue to neuroendocrine cells present in the intrahepatic bile duct epithelium [63,66].

Researchers have proposed that the NEC component within the HCC may be related to neuroendocrine cells within the primary HCC tumor as an integral part of histological constituents. Alternatively, this occurrence could be due to phenotypic alterations and/or divergent differentiation of HCC cells under specific conditions, or it might stem directly from hepatic stem cells with subsequent neuroendocrine differentiation [12]. Overall, the pathogenesis of MINEN tumors of the liver needs to be further elucidated. The concurrent occurrence of HCC and NEC in the liver is classified into “combined” or “collision” types based on histological patterns. In the combined MINEN, the two components closely intermingle within a single tumor nodule, whereas a collision tumor presents two distinct tumor areas composed of histologically different neoplastic cells [11,12,13,15]. In our case, a lesser HCC component was observed within the NEC component, with focal transition and blending into the NEC. A thin fibrous septum separated most HCC components, but a focal transition and blending was presented into the NEC. Therefore, the tumor was classified as a “combined” primary NEC and HCC of the liver. The rapid regrowth of liver tumoral lesions after TACE with the appearance of metastatic satellite lesions suggests a potential association with neuroendocrine components. Some authors have described a TACE-induced phenotypic change or the dedifferentiation of HCC [16]. Another possible explanation is the pre-existence of the neuroendocrine component, unresponsive to TACE and responsible for disease progression. Furthermore, HCC cells may undergo sarcomatous dedifferentiation, and TACE could induce or accelerate these sarcomatous changes in HCC through necrosis and degeneration [67,70]. Among the cases reported in the literature and described in this review, six patients had undergone previous transarterial chemoembolization (TACE) and one had received radiofrequency ablation (RFA). A case of hepatocarcinoma treated as part of a randomized phase III trial with tremelimumab (anti-CTLA-4) followed by durvalumab (anti-PD-L1) every 4 weeks is reported in the literature; in this case, biopsy taken after treatment showed the presence of neuroendocrine carcinoma; the authors conclude that it could represent a documented event of neuroendocrine transition in the context of HCC [71].

Beard R.E. et al. describe a case with triple components (HCC-NEC-CC) and performed a comparative molecular profiling between the different growth components of the tumor (neuroendocrine versus HCC-like) indicates that the derivation of these components was an early event in carcinogenesis. A lack of detectable neuroendocrine-associated mutations in the HCC-like component supports that the neuroendocrine component likely was not derived from the HCC-like cancer cells. More likely is that the early division of the growth components arose from a stem cell precursor capable of differentiation along separate cell lineage lines [50].

Given the rarity of this neoplasm, no standardized treatment regimen has been established. Therapeutic options include surgery, radiofrequency ablation, chemotherapy alone, or multimodal approaches. Treatment decisions should be made within a multidisciplinary framework. The aggressive nature of the disease, along with the poor response to therapy associated with the high-grade neuroendocrine component, highlights the critical importance of accurate diagnosis and personalized treatment strategies.

The cases identified in our literature review were managed using various treatment modalities. Surgery alone resulted in a median survival of 10 months; however, the most effective approach may be a combination of surgery and systemic therapy. Chemotherapy with platinum and etoposide was the most-used regimen. Mulsant et al. [47] reported a survival of 24 months in a patient treated with atezolizumab and bevacizumab, a regimen typically used for advanced hepatocellular carcinoma. This strategy could be supported by the hypothesis proposed by Yamaguchi et al. [12], suggesting that the neuroendocrine component arises from phenotypic alterations and/or the divergent differentiation of hepatocellular carcinoma cells under specific conditions. Alternatively, it may originate directly from hepatic stem cells undergoing subsequent neuroendocrine differentiation [10].

Multiphasic contrast-enhanced abdominal CT or nuclear magnetic resonance scans are recommended for establishing the diagnosis and staging of liver cancer. The choice of optimal imaging modality and contrast agent depends on various factors outlined in the Liver Imaging Reporting and Data System (LI-RADS) [72].

Lesions failing to meet imaging criteria necessitate a personalized approach, potentially involving additional imaging or biopsy guided by multidisciplinary discussions outlined in treatment protocols. Indications for core needle biopsy include cardiac cirrhosis, congenital hepatic fibrosis, or cirrhosis due to vascular disorders, such as the Budd–Chiari syndrome, hereditary hemorrhagic telangiectasia, or nodular regenerative hyperplasia, as well as elevated CA 19-9 or carcinoembryonic antigen levels suggestive of intrahepatic cholangiocarcinoma or “mixed” or “combined” HCC–cholangiocarcinoma. In addition to patients’ mild discomfort, liver biopsy carries inherent risks, such as seeding, and potential severe albeit rare complications, such as gallbladder perforation, bile peritonitis, haemobilia, pneumothorax, or hemothorax [73,74,75,76,77,78,79]. A meta-analysis on 1340 biopsies reported a tumor-seeding risk of 2.7%, with recent series reporting a risk of less than 1% [80,81,82,83]. However, these potential harms are balanced by the important benefits of liver nodule biopsy which include mitigating the risk of misdiagnosis, evaluating microscopic vascular invasion, low cost, reproducibility, potential for longitudinal review, prognostic stratification, and the identification of therapeutic targets [84].

## 5. Conclusions

Given the rarity of mixed neuroendocrine carcinoma (NEC) and hepatocellular carcinoma (HCC), our literature review is subject to inherent biases and limitations. There is considerable variability in the reported data in terms of demographics, treatments, and outcomes, with the latter probably related to the heterogeneity of clinical characteristics and treatment modalities. The lack of standardized treatment guidelines highlights the need for a multidisciplinary approach to develop personalized therapeutic strategies tailored to individual patient profiles. Additionally, biopsy remains essential for a more precise diagnostic assessment and a better understanding of the biological characteristics of primary hepatic tumors.

Comprehensive tumor profiling through advanced molecular and genetic analyses could provide valuable insights into the biological behavior of these mixed neoplasms, potentially paving the way for novel therapeutic approaches. Future studies should investigate the efficacy of emerging systemic therapies, including immune checkpoint inhibitors and targeted molecular agents, to improve clinical outcomes. Collaborative research efforts and international case registries could help define prognostic factors and optimize treatment strategies for this rare and aggressive malignancy. Furthermore, a deeper understanding of the mechanisms underlying neuroendocrine differentiation in hepatic malignancies may lead to innovative diagnostic and therapeutic strategies in this challenging clinical setting.

## Figures and Tables

**Figure 1 cancers-17-01074-f001:**
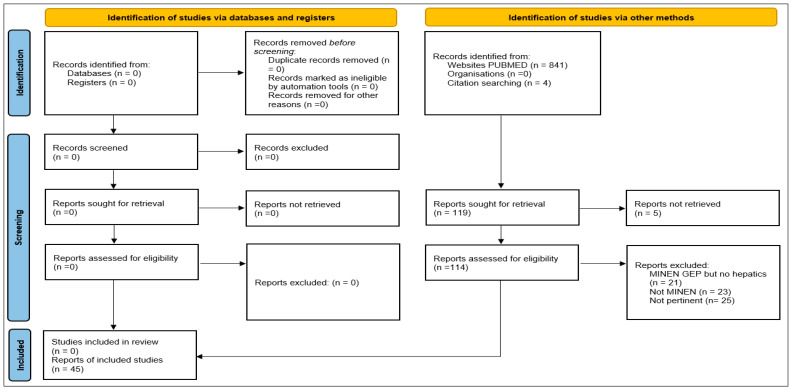
PRISMA 2020 flow diagram.

**Figure 2 cancers-17-01074-f002:**
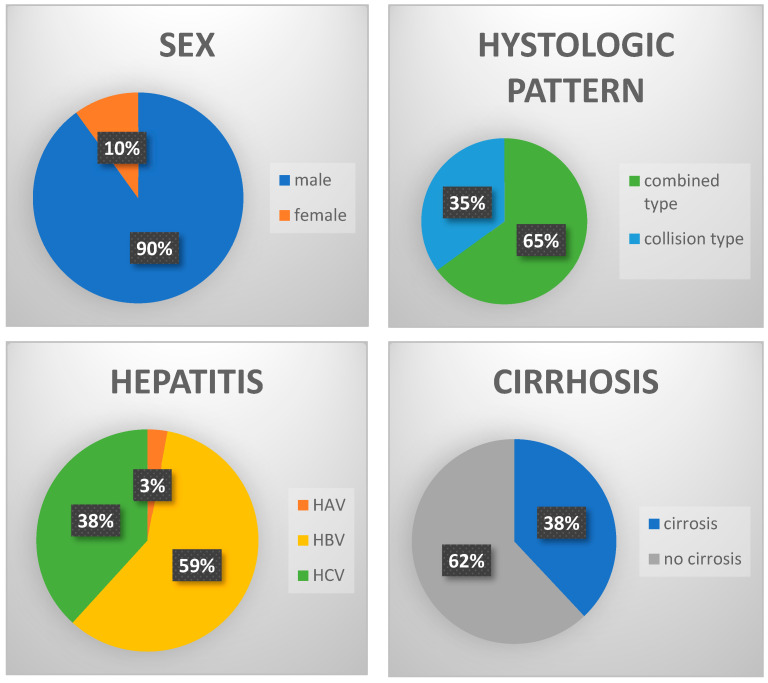
Distribution of sex, histology, hepatitis, and cirrhosis in cases reported in the literature.

**Figure 3 cancers-17-01074-f003:**
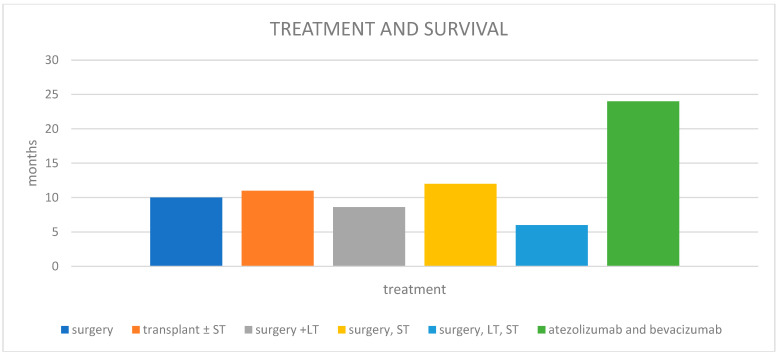
Treatment and median overall survival. LT: local treatment (PTPE: percutaneous transhepatic portal vein embolization; RFA: radiofrequency thermoablation; TACE: transarterial); ST: systemic treatment.

**Table 1 cancers-17-01074-t001:** Pertinent clinical cases.

Reference	Nationality	Age	Sex (M/F)	Histology	Type	Hepatitis	Cirrhosis (Yes/No)	Treatment	Survival from Diagnosis (Months)
Barsky S.H. et al., 1984 [8]	USA	43	M	HCC-NEN	combined	HBV	No	CT (Doxorubicin, Fluorouracil), Surgery	26
**Artopoulos J.G. et al., 1994** [9]	Greece	69	M	HCC-NEN	combined	HBV	No	Surgery	NR
**Vora I.M. et al., 2000** [10]	India	63	M	HCC-NEN	combined	/	Yes	Surgery	1
Ishida M. et al. 2003 [11]	Japan	72	M	HCC-NEN	collision	HCV	Yes	Surgery	NR
Yamaguchi R. et al., 2004 [12]	Japan	71	M	HCC-NEN	combined	HCV	No	Surgery	>5
**Garcia M.T. et al., 2006** [13]	USA	50	M	HCC-NEC	collision	HCV	No	Surgery, TACE, CT (Doxorubicin, Thalidomide, Bevacizumab)	>16
Yang C.S. et al., 2009 [14]	Taiwan	65	M	HCC-NEC	combined	HBV	No	Surgery	12
**Tazi E.M. et al., 2011** [15]	Morocco	68	M	HCC-NEC	collision	HBV	No	Surgery, CT (Cisplatin, Etoposide)	>28
Nakanishi C. et al., 2012 [16]	Japan	76	M	HCC-NEC	combined	HCV	No	TACE, Surgery	12
Hammedi F et al. 2012 [17]	Tunisia	51	M	HCC-NEN	combined	No	No	Palliative	1
Aboelenen A. et al., 2014 [18]	Egypt	51	M	HCC-NEN	combined	HCV	No	Surgery	>6
Choi G.H. et al., 2016 [19]	South Korea	72	M	HCC-NEC	collision	HCV	No	Surgery, Liver CT, CT (Cisplatin, Etoposide)	>10
Baker E. et al., 2016 [20]	USA	76	M	HCC-NEC	collision	No	Yes	Surgery, CT (Platinum Based)	/
Nishino H. et al., 2016 [21]	Japan	72	M	HCC-NEC	combined	HCV	No	Surgery, CT	3
Yun E.Y. et al., 2016 [22]	Korea	/	/	HCC-NEC		HBV	Yes	/	/
Matsumoto H. et al., 2017 [23]	Japan	77	M	HCC-NEC		/	/	Surgery	/
Nomura Y. et al., 2017 [24]	Japan	71	M	HCC-NEC	combined	HCV	No	Surgery	10
		71	M	HCC-NEC	collision	HCV	No	RFA, Surgery	8.6
		58	M	HCC-NEC	combined	HBV	No	Surgery	>19.4
		50	M	HCC-NEC	combined	HBV	Yes	Surgery	>19.3
		63	M	HCC-NEC	combined	HCV	No	Surgery	>23.7
Liu Y.J et al., 2017 [25]	Taiwan	65	M	HCC-NEC	collision	HCV	/	Surgery	1.3
Okumura Y. et al., 2017 [26]	Japan	70	M	HCC-NEC	collision	/	/	TACE and PTPE, Surgery, CT	3
Chuah K.L. et al., 2018 [27]	China	77	M	HCC-NEC		HBV	/	Palliative	/
**Kwon H.J. et al., 2018** [28]	Korea	44	M	HCC-NEC	combined	HBV	Yes	Surgery	2
Yilmaz D.B. et al., 2018 [29]	Turkey	56	M	HCC-NEC	collision	No	Yes	Surgery Transplantation	>10
Jahan N. et al., 2020 [30]	USA	50	M	HCC-NEC	combined	HCV	Yes	Surgery, CT	33
Mita H. et al., 2021 [31]	Japan	81	F	HCC-NEC		No	Yes	NR	/
Wang H. et al., 2021 [32]	China	33	M	HCC-NEC		HBV	Yes	Surgery	>6
Ikeda A. et al. 2021 [33]	Japan	79	M	HCC-NEC	combined	No	No	Surgery	4
Nakano A. et al., 2021 [34]	Japan	84	F	HCC-NEC	combined	No	No	Surgery	>9
Lan J. et al., 2021 [35]	China	39	M	HCC-NEN	/	HBV	No	TACE, Surgery, CT (Cisplatin, Etoposide)	6
Khanam R. et al., 2021 [36]	USA	66	F	HCC-NEN	/	No	No	CT, Immunotherapy	NR
Shi C. et al., 2021 [37]	USA	/	/	HCC-NEC	/	/	Yes	/	/
**Jeng K.S. et al., 2022** [38]	Taiwan	48	M	HCC-NEC	collision	HBV	No	TACE/RFA	2
Noh B.G. et al., 2022 [39]	Korea	73	M	HCC-NEC	collision	HBV	No	Surgery	>24
Tanaka H. et al., 2023 [40]	Japan	70	M	HCC-NEC	mixed	HBV	No	Surgery	>12
Gao X. et al., 2023 [41]	China	58	M	HCC-LCN	/	HBV	Yes	Surgery, CT (Cisplatin, Etoposide)	>8
Ahmed Z. et al., 2023 [42]	USA	67	M	HCC-NEC	combined	No	Yes	TACE, RFA	NR
**Ben Kridis W. et al., 2023** [43]	Tunisia	44	M	HCC-NEN	/	No	No	Palliative	1
Shin W.Y. et al., 2023 [44]	Korea	63	M	HCC-NEN	combined	HBV	Yes	Surgery, CT (Cisplatin, Etoposide)	12
Tsuji K. Et al., 2024 [45]	Japan	70	M	HCC-NEC	/	HBV	/	Surgery	11
Mubeen B. et al., 2024 [46]	India	73	M	HCC-NEC	/	No	No	Surgery, Palliative RT of Adrenal Metastasis	>14
	India	44	F	HCC-NEC	combined	No	Yes	Transplant, Adjuvant Cisplatin Etoposide	>12
Mulsant M. et al., 2025 [47]	France	/	NR	HCC-NEC (MEN 1)		/	/	Atezolizumab+ Bevacizumab Lenvatinib Cabozantinib	24
Bellavia S, personal communication, 2025	Italy	76	F	HCC-NEC	combined	HAV	Yes	TACE, Surgery	7
He C et al., 2013 [48]	China	57.5	M	HCC-NEC-CC	/	HBV	/	/	/
			M	HCC-NEC-CC	/	HBV	/	/	/
Kano Y. et al., 2014 [49]	Japan	73	M	NEC-CC metachronous HCC	/	/	/	Surgery	/
Beard R.E. et al., 2017 [50]	USA	19	M	HCC-NEC-CC	/	/	/	Surgery, Capecitabine Temozolomide	>8
**Dimopoulos Y.P. et al. 2021** [51]	USA	65	F	HCC-NEC-CC		HBC	/	Surgery, CT	12
Wu Y. et al., 2022 [52]	China	32	M	HCC-NEC-CC		HBV	/	Surgery, TACE	3

M: male; F: female; HCC: hepatocellular carcinoma; NEC: neuroendocrine carcinoma; CC: cholangiocarcinoma; LCN: large-cell neuroendocrine; HBV: hepatitis B virus; HCV: hepatitis C virus; CT: chemotherapy; PTPE: percutaneous transhepatic portal vein embolization; RT radiotherapy; TACE: transarterial chemoembolization; RFA: radiofrequency thermoablation; NR: not reported.

**Table 2 cancers-17-01074-t002:** Treatment modality.

Author	Treatment Modality	Median Survival Month (Range)
[9,10,11,12,14,18,23,24,28,32,33,34,39,40,45,46]	Surgery	10 (1–24)
[9,11,23]		NR
[29]	Transplant	10
[46]	Transplant, Adjuvant CT	12
[8]	CT, Surgery	26
[15,19,20,21,30,41,44]	Surgery, CT	11 (8–33)
[20]		NR
[13]	Surgery, TACE, CT	16
[24]	RFA, Surgery	8.6
[16], Bellavia personal communication	TACE, Surgery	9.5 (7–12)
[35]	TACE, Surgery, CT	6
[26]	TACE and PTPE, Surgery, CT	3
[38]	TACE, RFA	2
[42]		NR
[47]	Atezolizumab and Bevacizumab, then Lenvatinib and Cabozantinib	24
[36]	CT, Immunotherapy	NR
[17,23,43]	Palliative	1
[27]		NR

CT: chemotherapy; PTPE: percutaneous transhepatic portal vein embolization; RFA: radiofrequency thermoablation; TACE: transarterial chemoembolization; NR: not reported.

## Data Availability

Dataset available on request from the authors.

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
