# Peer review of "Combined Neuroendocrine Carcinoma and Hepatocellular Carcinoma of the Liver: Systematic Literature Review Suggests Implementing Biological Characterization to Optimize Therapeutic Strategy"

_cancers, 2025, doi:10.3390/cancers17071074_

Round 1
Reviewer 1 Report
Comments and Suggestions for Authors
The article describes and systematically revises the case reports concerning this rare diseases. The authors should be commended for gathering such amount of information.
THe authors stated that the median overall survival was 10 months. How did they calculate this outcome? Just taking a look at Table 1, median overall survival doesn't seem 10 months....
The authors should comment more in the discussion about the impact of NEN on the liver (cite the series PMID: 27956320)
Which systemic therapies could be used in this kind of mixed neoplasia? COmment on the rationale to the current evidence
Author Response
For research article
|
Response to Reviewer X Comments
|
||
|
1. Summary |
|
|
|
Thank you very much for taking the time to review this manuscript. Please find the detailed responses below and the corresponding revisions/corrections highlighted/in track changes in the re-submitted files
|
||
|
2. Questions for General Evaluation |
Reviewer’s Evaluation |
Response and Revisions |
|
Does the introduction provide sufficient background and include all relevant references? |
Can be improved |
We implemented the introduction as required by the Referees
|
|
Are all the cited references relevant to the research? |
Yes |
|
|
Is the research design appropriate? |
Yes |
|
|
Are the methods adequately described? |
Yes |
|
|
Are the results clearly presented? |
Yes |
|
|
Are the conclusions supported by the results?
|
Yes |
|
|
3. Point-by-point response to Comments and Suggestions for Authors |
||
|
1. Comments 1: The article describes and systematically revises the case reports concerning this rare diseases. The authors should be commended for gathering such amount of information. |
||
|
Response 1: Thank you very much for taking the time to review this manuscript and for this appreciation
|
||
|
1. Comments 2: The authors stated that the median overall survival was 10 months. How did they calculate this outcome? Just taking a look at Table 1, median overall survival doesn't seem 10 months...
|
||
|
Response 2: We used a statistical software for calculations, repeated them and the median is 10 months. 1. Comment 3: the authors should comment more in the discussion about the impact of NEN on the liver (cite the series PMID: 27956320) Response 3: We added in the discussion a paragraph with a brief description about the treatment and cited the suggested article.
1. Comment 4: Which systemic therapies could be used in this kind of mixed neoplasia? Comment on the rationale to the current evidence. Response 4: We added in the discussion a paragraph about the treatments with the best survival found in published cases and suggested a treatment hypothesis such as the combination of atezolizumab bevacizumab, standard treatment for advanced hepatocarcinoma, with the rationale that suggested this hypothesis to us.
|
||
|
4. Response to Comments on the Quality of English Language |
||
|
Point 1: The English is fine and does not require any improvement.
|
||
|
Response 1: |
||
|
5. Additional clarifications |
||
none

Reviewer 2 Report
Comments and Suggestions for Authors
Cancers (Manuscript ID: cancers-3492406), Comments to the Authors:
Title: Combined neuroendocrine carcinoma and hepatocellular carcinoma of the liver: Systematic literature review
Comments
The submitted review highlighted cases of primary mixed HCC and NEC in the liver. The search adhered to PRISMA guidelines, and relevant studies were critically analyzed. A total of 45 documented cases were reviewed, focusing on patient demographics, clinical characteristics, treatment strategies, and outcomes. Most patients (90%) were male, with a median age of 66.5 years. Hepatitis B or C infection was present in 74% of cases, and liver cirrhosis was reported in 38%. The combined type was the most frequently observed histological pattern (65%). Treatment modalities varied, including transarterial chemoembolization (TACE), radiofrequency ablation (RFA), surgery, and systemic therapies. The median overall survival was 10 months, highlighting the aggressive nature of these tumors.
I think the submitted review can be accepted after the authors respond to the following comments:
- The title is confusing, the authors should rephrase the title to provide more information about the work in the submitted paper.
- There are no graphs in the submitted review. The authors may try to summarize their findings in an illustrative figure.
- By reviewing table 1, I found many missing data points that will definitely affect the results obtained by the authors. How did the authors handle this issue.
- The methods lack detail on statistical approaches, particularly for handling censored survival data. Can the authors comment on this issue?
- Heterogeneity was very common in many cases in terms of demographics, treatments, and outcomes. How did the authors handled this issue and did it affect the conclusion.
- The authors should include “limitations” subsection in their review.
- The authors should expand the discussion on molecular mechanisms and therapeutic prospects beyond general statements.
The English quality is fine.
Author Response
For research article
|
Response to Reviewer X Comments
|
||
|
1. Summary |
|
|
|
Thank you very much for taking the time to review this manuscript. Please find the detailed responses below and the corresponding revisions/corrections highlighted/in track changes in the re-submitted files.
|
||
|
2. Questions for General Evaluation |
Reviewer’s Evaluation |
Response and Revisions |
|
Does the introduction provide sufficient background and include all relevant references? |
Can be improved |
we improved the introduction |
|
Are all the cited references relevant to the research? |
Can be improved |
We believe the references cited are necessary
|
|
Is the research design appropriate? |
Can be improved |
we improved the research design with all the suggestions received from the reviewers
|
|
Are the methods adequately described? |
Can be improved |
we improved the description of methods
|
|
Are the results clearly presented? |
Can be improved |
we improved the results with figures
|
|
Are the conclusions supported by the results? |
Can be improved |
we modified the conclusion
|
|
3. Point-by-point response to Comments and Suggestions for Authors |
||
|
||
|
Response 1: Thank you for pointing this out. We agree with this comment. Therefore, we have changed the title: “COMBINED NEUROENDOCRINE CARCINOMA AND HEPATOCELLULAR CARCINOMA OF THE LIVER: SYSTEMATIC LITERATURE REVIEW SUGGESTS IMPLEMENTING BIOLOGICAL CHARACTERIZATION TO OPTIMIZE THERAPEUTIC STRATEGY”. |
||
|
||
|
Response 2: Agree. We have add illustrative figures of results
Response 3: Data missing from the table were not shown in the individual reports. The analyses reported are only percentages and medians that refer to cases where the data was present.
1. Comment 4: The methods lack detail on statistical approaches, particularly for handling censored survival data. Can the authors comment on this issue? Response 4: We added in the materials and methods, For the description of results, we used percentages and medians with range using only known data.
Response 5: There is considerable heterogeneity in the reported data, which mainly relates to treatment modalities, so we limited ourselves to case descriptions, and medians were reported in the table along with the number of cases where survival was reported.
1. Comment 6: The authors should include “limitations” subsection in their review. Response 6: In the “conclusions section” we added that the reported results have limitations: There is considerable variability in the reported data in terms of demographics, treatments, and outcomes, the latter probably related to the heterogeneity of clinical characteristics and treatment modalities.
1. Comment 7: The authors should expand the discussion on molecular mechanisms and therapeutic prospects beyond general statements. Response 7: in the discussion we have reported a further paragraph concerning treatment.
|
||
|
4. Response to Comments on the Quality of English Language |
||
|
Point 1: The English could be improved to more clearly express the research. |
||
|
Response 1: we improved the English |
||
|
5. Additional clarifications |
||
|
none |
||

Reviewer 3 Report
Comments and Suggestions for Authors
The authors present a research paper based on previous studies on the combined cancers in liver and draw the conclusions. But i dont find any new data being generated nor are there any strong conclusions drawn from this study. I would suggest the authors to convert it into a review and present some future perspective in this area.
Author Response
For research article
|
Response to Reviewer X Comments
|
||
|
1. Summary |
|
|
|
Thank you very much for taking the time to review this manuscript. Please find the detailed responses below and the corresponding revisions/corrections highlighted/in track changes in the re-submitted files. [This is only a recommended summary. Please feel free to adjust it. We do suggest maintaining a neutral tone and thanking the reviewers for their contribution although the comments may be negative or off-target. If you disagree with the reviewer's comments please include any concerns you may have in the letter to the Academic Editor.]
|
||
|
2. Questions for General Evaluation |
Reviewer’s Evaluation |
Response and Revisions |
|
Does the introduction provide sufficient background and include all relevant references? |
Yes |
|
|
Are all the cited references relevant to the research? |
Yes |
|
|
Is the research design appropriate? |
Yes |
|
|
Are the methods adequately described? |
Yes |
|
|
Are the results clearly presented? |
Yes |
|
|
Are the conclusions supported by the results? |
Yes |
|
|
3. Point-by-point response to Comments and Suggestions for Authors |
||
|
Comments 1: The authors present a research paper based on previous studies on the combined cancers in liver and draw the conclusions. But i dont find any new data being generated nor are there any strong conclusions drawn from this study. I would suggest the authors to convert it into a review and present some future perspective in this area |
||
|
|
||
|
Response 1: We have modified and added some paragraphs regarding the possible conclusions in this literature review that concerns only a few cases. The discussion and conclusions have been modified to be able to underline the evidence that can be useful and applicable in the future and that concern the need for a better biological definition with the implementation of liver biopsies and a therapeutic suggestion, with relative rationale, that is reported in a literature case.
|
||
|
|
||
|
4. Response to Comments on the Quality of English Language |
||
|
Point 1: The English is fine and does not require any improvement. |
||
|
|
||
|
5. Additional clarifications |
||
|
None |
||

Round 2
Reviewer 1 Report
Comments and Suggestions for Authors
The manuscript is OK
Reviewer 2 Report
Comments and Suggestions for Authors
Cancers (Manuscript ID: cancers-3492406), Comments to the Authors:
Title: Combined neuroendocrine carcinoma and hepatocellular carcinoma of the liver: Systematic literature review
Comments
After reading the authors response to my comments, I think the revised manuscript can be accepted for publication.
Comments on the Quality of English LanguageThe English language of the paper is fine.
Reviewer 3 Report
Comments and Suggestions for Authors
accept the revised manuscript